# Severity by National Institute of Health Stroke Scale Score and Clinical Features of Stroke Patients with Patent Foramen Ovale Stroke and Atrial Fibrillation

**DOI:** 10.3390/jcm10020332

**Published:** 2021-01-18

**Authors:** Kaito Abe, Fumiya Hasegawa, Ryota Nakajima, Hidetoshi Fukui, Moto Shimada, Takahiro Miyazaki, Hiroshi Doi, Goro Endo, Kaori Kanbara, Yasuyuki Mochida, Jun Okuda, Nobuya Maeda, Akira Isoshima, Koichi Tamura, Tomoaki Ishigami

**Affiliations:** 1Department of Cardiology, Omori Red Cross Hospital, 4-30-1 Chuo, Ota-Ward, Tokyo 143-8527, Japan; kaitoabemed@gmail.com (K.A.); e103061a@gmail.com (F.H.); cisvjikoryota@gmail.com (R.N.); fukubobu777@yahoo.co.jp (H.F.); petapetapeta@live.jp (M.S.); taka.miyazaki.55@gmail.com (T.M.); doihiroshi6120@yahoo.co.jp (H.D.); goro.08.29.09054261798@docomo.ne.jp (G.E.); kkanbara.circ@gmail.com (K.K.); yasmochi90@gmail.com (Y.M.); jokudajun@yahoo.co.jp (J.O.); 2Department of Medical Science and Cardiorenal Medicine, Yokohama City University, 3-9 Fukuura Kanazawa-Ward, Yokohama City 236-0004, Japan; tamukou@yokohama-cu.ac.jp; 3Department of Neurology, Omori Red Cross Hospital, 4-30-1 Chuo, Ota-Ward, Tokyo 143-8527, Japan; nobmaeda@khf.biglobe.ne.jp; 4Department of Neurosurgery, Omori Red Cross Hospital, 4-30-1 Chuo, Ota-Ward, Tokyo 143-8527, Japan; a-isoshima@omori.jrc.or.jp

**Keywords:** patent foramen ovale and stroke, atrial fibrillation and stroke, cryptogenic stroke, severity of stroke, National Institute of Health Stroke Scale score

## Abstract

The comparative severity of patent foramen ovale (PFO)-related stroke in patients without atrial fibrillation (AF) and AF-related stroke in patients without PFO is unknown. Therefore, we compared the severity of PFO-related stroke and AF-related stroke. Twenty-six patients who underwent transesophageal echocardiography (TEE) were diagnosed with cardioembolic stroke from July 2018 to March 2020. Cases with AF detected by electrocardiograms or thrombus in the left atrium or left atrial appendage on TEE were included in the AF-related stroke group. Cases with a positive microbubble test on the Valsalva maneuver during TEE, and with no other factors that could cause stroke, were included in the PFO-related stroke group. This study was designed as a single-center, small population pilot study. The stroke severity of the two groups by the National Institute of Health Stroke Scale (NIHSS) score was compared by statistical analysis. Of the 26 cases, five PFO-related stroke patients and 21 AF-related stroke patients were analyzed. The NIHSS score was 2.2 ± 2.8 and 11.5 ± 9.2 (*p*-value < 0.01), the rate of hypertension was 20.0% and 85.7% (*p*-value = 0.01), and the HbA1c value was 5.5 ± 0.2% and 6.3 ± 1.3% (*p*-value = 0.02) in the PFO-related and AF-related stroke groups, respectively. Compared with AF-related stroke patients, stroke severity was low in PFO-related stroke patients.

## 1. Introduction

Stroke results in substantial disability and sometimes causes death [1]. The TOAST classification denotes five subtypes of ischemic stroke: (1) large-artery atherosclerosis, (2) cardioembolism, (3) small-vessel occlusion, (4) stroke of other determined etiology, and (5) stroke of undetermined etiology [2]. Cardioembolic stroke accounts for 15–30% of ischemic strokes [3].

In up to 40% of patients with acute ischemic stroke, there is a stroke of undetermined etiology in TOAST classification (5); this stroke has been labeled as cryptogenic stroke [1,2,4,5]. Major cardioembolic risk sources include atrial fibrillation (AF), recent myocardial infarction, previous myocardial infarction (left ventricular aneurysm), intracardiac thrombus, tumors, rheumatic valve disease, aortic arch atheromatous plaques, endocarditis, and mechanical valve prosthesis, whereas minor or unclear risk sources include patent foramen ovale (PFO), atrial septal aneurysm (ASA), and giant Lambl’s excrescences [3]. Evaluation of stroke sources is important for preventing second stroke events.

In the general population, 0.4–1% have AF, and the prevalence increases to 9% in the population aged 80 years or older [6]. The CHADS2 and CHA2DS2-VASc risk scores show the frequent occurrence of stroke and embolism, ranging from 0 (low risk) to 18% event/year (high-risk) among patients with AF [7,8].

For the management of AF, anticoagulant therapy, catheter ablation, and antiarrhythmic drugs are well-established [9]. Recently, transcatheter left atrial appendage closure has been used as a primary therapy for AF patients with contraindications for using chronic oral anticoagulation to prevent stroke [10].

PFO is caused by incomplete fusion of the septum primum and secundum after birth in the cranial portion of the fossa ovalis, and is a common anatomical variant found in about 25% of the general population [11,12]. Stroke with PFO occurs when a systemic venous thrombus travels directly into the systemic arterial circulation [1]. The proportion of stroke patients with PFO is 21–63% [11]. According to a report, cryptogenic stroke patients with PFO were younger and less likely to have conventional vascular risk factors than cryptogenic stroke patients without PFO [11].

Recently, the DEFENSE, REDUCE, and CLOSE trials demonstrated the superiority of PFO closure over medical management [13,14,15]. In cryptogenic stroke, detection of PFO is important to select an adequate secondary stroke prevention therapy. Transesophageal echocardiography (TEE) is the gold standard for PFO detection. The microbubble test with Valsalva maneuver is recommended for detecting PFO on TEE to avoid the increasing false negative rate of up to 20% when the Valsalva maneuver is not performed [3,16].

It is known that the severity of ischemic stroke patients with PFO, including patients with AF, is lower than that of patients without PFO, including patients with AF [17]; however, whether the severity of PFO-related stroke in patients without AF is lower than that of AF-related stroke in patients without PFO is unknown. Thus, the purpose of this analysis was to evaluate the severity of PFO-related stroke and AF-related stroke, and to identify the characteristics of both stroke types.

## 2. Materials and Methods

### 2.1. Study Design and Patient Population

We performed a single-center (Omori Red Cross Hospital) retrospective study on consecutive patients with cardioembolic stroke, including suspected cases on magnetic resonance imaging, who underwent TEE between July 2018 and March 2020.

Patients with AF diagnosed from history, electrocardiogram (ECG) at admission, 24 h-holter ECG monitoring, ECG monitoring in the ward, or patients with thrombi including smoke-like echo with a swirling motion of blood in the left atrium (LA) or left atrial appendage (LAA), which is known to be a marker of a prothrombotic state, were classified as AF-related stroke patients [18]. Patients with PFO without AF were classified as PFO-related stroke patients.

Patients with mobile aortic plaque were defined as Class V in the Katz Index [19], patients after valve replacement and those with cardiac tumor, infectious endocarditis, Lambl’s excrescence on the aortic valve, or those diagnosed with atherosclerotic stroke by neurologists after TEE were excluded from this study. Patients with PFO between the right atrium (RA) and LA diagnosed only by the color Doppler method without passage of microbubbles were excluded from this study.

### 2.2. Evaluations

The diagnosis of ischemic stroke was made by neurologists with known experience in cerebrovascular diseases. TEE was performed and evaluated by cardiologists who were well-experienced in echocardiology. TEE was performed with either Vivid E95 (GE Healthcare, Tokyo, Japan) or ALOKA Prosoundα10 (ALOKA, Tokyo, Japan).

### 2.3. Baseline Study Assessment

We collected data on patient characteristics (age, sex, height, body weight, and smoking habit); vascular risk factors; the administration ratio of antiplatelet therapy or oral anticoagulant therapy before stroke onset; blood tests (aspartate transaminase (AST), alanine aminotransferase (ALT), lactate dehydrogenase (LDH), alkaline phosphatase (ALP), total bilirubin (T-bil), brain natriuretic peptide (BNP), hemoglobin, HbA1c, D-dimer, low-density lipoprotein (LDL), high-density lipoprotein (HDL), triglycerides (TGs), creatinine (Cre), estimated glomerular filtration rate (eGFR), and creatinine clearance (CCR) (Cockcroft–Gault equation)); HAS-BLED score to assess bleeding risk in AF patients from hypertension, abnormal renal/liver function, stroke, bleeding history or predisposition, labile INR, elderly (>65 years old), and concomitant drugs/alcohol use [20]; National Institute of Health Stroke Scale (NIHSS) [21]; the risk of paradoxical embolism (RoPE) score to assess the likelihood of the cryptogenic stroke being related to PFO based on the scoring items of age, hypertension, diabetes, history of stroke or transient ischemic attack, smoking habit, and cortical infarct on imaging [11]; ECG; 24 h-holter ECG monitoring; ECG monitoring in the hospital ward; thrombus, including smoke-like echo in LA or LAA by TEE; and left ventricular ejection fraction (LVEF) by transthoracic echocardiography.

The TEE was performed under light sedation with propofol. The LA or LAA thrombi were evaluated in all patients by TEE. For all patients who underwent TEE, an intravenous microbubble test during the Valsalva maneuver was performed. In the GORE-REDUCE trial [14], the classification of PFO size was based on the maximum number of microbubbles during the first three cardiac cycles; 0 microbubbles were classified as no shunt, one to five microbubbles as small, six to 25 microbubbles as moderate, and more than 25 microbubbles as large. PFO was diagnosed by the microbubble test using the Valsalva maneuver technique between the RA and LA. Complex PFO was classified as PFO with ASA, or with a long tunnel length of over 8 mm, or with the eustachian valve [22]. Some AF-related stroke patients with LA or LAA thrombi skipped the microbubble test.

### 2.4. Statistical Analysis

JMP Pro version 15 software (SAS Institute Japan Inc., Tokyo, Japan) was used for statistical analysis. Data are expressed as mean ± standard deviation for continuous variables and as frequencies and percentages for categorical variables. Baseline characteristics were compared using the Student’s *t* test or Welch’s *t* test for continuous variables and Fisher’s exact test for categorical variables. A *p*-value < 0.05 was considered statistically significant.

## 3. Results

### 3.1. Stroke Classification

A total of 82 patients were enrolled from July 2018 to March 2020. AF-related stroke was noted in 21 (25.6%) patients, atherosclerotic stroke in 21 (25.6%), PFO-related stroke in five (6.1%), cardioembolic stroke without AF in 13 (15.9%), others (systemic lupus erythematosus, hyperemia, vasculitis, lacunar infarction) in four (4.9%), and cryptogenic stroke in 18 (22.0%) (Figure 1). Cardioembolic stroke without AF included patients with post-valve replacement (four patients), cardiac tumor (two patients), infectious endocarditis (one patient), Lambl’s excrescence on the aortic valve (two patients), old myocardial infarction (one patient), and patients with PFO between the RA and LA diagnosed only by the color Doppler method without passage of microbubbles (three patients).

### 3.2. Study Population and Patient Features

Among the 21 AF-related stroke patients, 20 (95.2%) patients had AF and 10 (47.6%) had LA or LAA thrombus, whereas the PFO-related stroke patients had no AF or thrombus in the LA or LAA. The ratio of comorbidity with hypertension in AF-related stroke patients was higher than that in PFO-related stroke patients (85.7%, 20.0%, *p*-value = 0.01). The NIHSS score in AF-related stroke patients was more severe than that in PFO-related stroke patients (11.5 ± 9.2, 2.2 ± 2.8, *p*-value < 0.01). Age, height, body weight, LVEF, and HAS-BLED score were not significantly different between the groups. The comorbidity ratio of dyslipidemia, diabetes, old myocardial infarction, or past stroke history had no significant differences between the two groups. The administration ratio of antiplatelet therapy or oral anticoagulant therapy before stroke onset had no significant differences between the two groups (Table 1).

HbA1c values in AF-related stroke patients were higher than that in PFO-related stroke patients (6.3% ± 1.3%, 5.5% ± 0.2%, *p*-value = 0.02). BNP, D-dimer, hemoglobin, LDL, HDL, TG, Cre, eGFR, CCR, AST, ALT, T-bil, and ALP were not significantly different between the groups (Table 2).

### 3.3. PFO Characteristics of This Study

Among the five PFO-related stroke patients, one patient had a small shunt PFO, two patients had moderate shunt PFO, and two patients had a large shunt PFO. Of the five PFO-related stroke patients, four patients had complex PFO; all complex PFOs had a long PFO tunnel, and had no eustachian valve or ASA. The RoPE score was 4.8 ± 2.4, the average PFO attributable fraction was 36.0 ± 37.3%, and the estimated two years stroke recurrence rate was 12.2 ± 7.2% in this PFO population by RoPE score to assess the likelihood of the cryptogenic stroke being related to PFO (Table 3) [11].

## 4. Discussion

The novelty of this study is that it is a PFO group diagnosed by TEE with the highly diagnostic Valsalva maneuver technique, and that it includes the severity of stroke between PFO-related stroke in patients without AF and AF-related stroke in patients without PFO.

Severity comparison by the NIHSS showed that AF-related stroke was more severe than PFO-related stroke. Previous studies [17] also reported that PFO-related stroke was less severe than strokes in the other groups; it was expected that AF-related stroke would be more severe, which was also revealed in our study. Regarding administration of oral medication before stroke onset, antiplatelet therapy was administered to two (40%) patients in the PFO-related stroke group. In the AF-related stroke group, anticoagulation therapy was administered to four (19.0%) patients and antiplatelet therapy to seven (33.3%). It may be necessary to consider the effect of these pre-medications on the severity of stroke.

Generally, it is said that PFO-related stroke patients are younger and have less cardiovascular risk; this study also showed that the ratio of comorbidity of hypertension and high HbA1c in blood tests was significantly lower in the PFO-related stroke patients compared to AF-related stroke patients. Although there was no significant difference in other factors, the age, ratio of comorbidity of dyslipidemia, diabetes, and ratio of smoking habits tended to be low in the PFO-related stroke group, which was similar to the existing report [11]. In particular, the average age of the PFO-related stroke patients was 58.2 years, which is younger than that of the AF-related stroke patients (77.9 years), and it is necessary to consider that the fact that there are few cardiovascular risk factors may also contribute to the low severity of stroke in the PFO-related stroke patients. For younger patients, functional improvement after stroke can be expected, and observation of long prognosis is also important.

It has been reported that the probability of stroke due to PFO is 88%, and the recurrence rate after two years is 2% in the group with the highest RoPE Score [11]. The PFO attributable stroke rate in this study is not high at 36.0%, but the average recurrence probability after two years by the RoPE score is estimated to be high at 12.2%. Therefore, secondary prevention therapies, such as antiplatelet therapy, anticoagulant therapy, or a PFO occluder device, seemed to be important in decreasing the recurrence rate of strokes.

This study has some limitations. This study is a pilot study not registered in clinical trials with an international clinical trials register. As this was a retrospective analysis at a single institution, it is necessary to consider the influence of the small population, five PFO-related stroke patients and 21 AF-related stroke patients. Statistical results may not be sufficient, as power analysis has not been performed in this study. It is also necessary to consider the influence of population bias, such as pre-test probability, because it is a group of cases in which neurologists suspected cardioembolic stroke and who needed TEE. In addition, there is a possibility that latent AF and PFO may be involved in cases of stroke other than cardioembolic stroke diagnosed by neurologists. Since patient prognosis was not observed in this study, it is necessary to observe their prognosis after medical interventions. We need to be careful in interpreting the results of this study.

There were 18 (22.0%) cases in the stroke group with unknown causes in the analysis target, and four of them were suspected of being cardioembolic stroke by the neurologist, but definitive clinical findings were unclear. It seems necessary to evaluate the comorbidity of AF with an implantable electrocardiograph.

In this analysis, TEE with the Valsalva maneuver technique was performed for PFO detection; therefore, PFO detection in this study was highly credible. However, it is said that there are detection limits of this method; hence, it seems that there is a possibility of wrongly classifying other strokes into the cryptogenic stroke group.

In addition, even in the group diagnosed with cryptogenic stroke, the detection rate of thrombus in LA/LAA may have decreased, because at the time of TEE, antiplatelet therapy or anticoagulant therapy had already been performed. In this study, after stroke, anticoagulant therapy was administered to three patients, and antiplatelet therapy was administered to one patient in the PFO-related stroke group. In the AF-related stroke group, anticoagulant therapy was administered to all patients. Of these, anticoagulation therapy alone was administered to 19 (90.5%) patients, and a combination of antiplatelet therapy and anticoagulant therapy was administered to two (9.5%) patients. Therefore, it is necessary to observe the secondary preventive effect on the recurrence rate in the future.

## 5. Conclusions

Compared with AF-related stroke patients, stroke severity, the comorbidity of hypertension rate, and the value of HbA1c were low in PFO-related stroke patients.

## Figures and Tables

**Figure 1 jcm-10-00332-f001:**
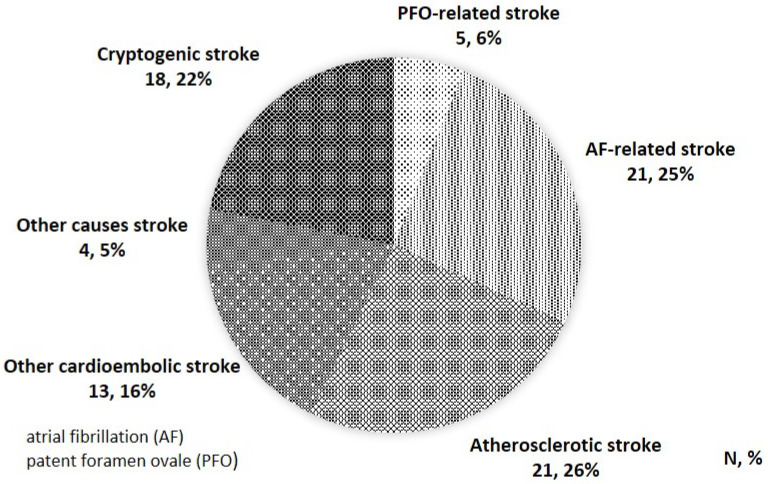
Stroke classification of this study.

**Table 1 jcm-10-00332-t001:** Baseline patient characteristics of PFO-related stoke and AF-related stroke.

Characteristics		PFO-Related Stroke	AF-Related Stroke	*p*-Value
		(*n* = 5)	(*n* = 21)	
Age	y.o	58.2 ± 23.4	77.9 ± 7.6	n.s *
Sex (Male)	*n* (%)	4 (80%)	14 (66.7%)	n.s †
Height	cm	166.4 ± 7.3	162.0 ± 8.0	n.s
Body Weight	kg	59.9 ± 10.7	58.9 ± 11.0	n.s
Smoking Habit	*n* (%)	5 (100%)	11 (52.4%)	n.s †
Hypertension	*n* (%)	1 (20%)	18 (85.7%)	0.01 †
Dyslipidemia	*n* (%)	3 (60%)	11 (52.4%)	n.s †
Diabetes	*n* (%)	0 (0%)	7 (33.3%)	n.s †
OMI	*n* (%)	0 (0%)	2 (9.5%)	n.s †
History of Stroke	*n* (%)	0 (0%)	2 (9.5%)	n.s †
PFO	*n* (%)	5 (100%)	0 (0%)	<0.01 †
AF	*n* (%)	0 (0%)	20 (95.2%)	<0.01 †
LA/LAA thrombus	*n* (%)	0 (0%)	10 (47.6%)	n.s †
LVEF	%	69.0 ± 5.2	66.0 ± 8.4	n.s
Antiplatet Therapy	*n* (%)	2 (40%)	7 (33.3%)	n.s †
Anticoagulant Therapy	*n* (%)	0 (0%)	4 (19%)	
DOAC	*n* (%)		3 (14.3%)	
Warfarin	*n* (%)		1 (4.8%)	
HAS-BLED Score		2.2 ± 1.6	3.3 ± 0.9	n.s
NIHSS Score		2.2 ± 2.8	11.5 ± 9.2	<0.01 *

Atrial fibrillation (AF), direct oral anticoagulants (DOAC), left atrium (LA), left atrial appendage (LAA), left ventricular ejection fraction (LVEF), National Institute of Health Stroke Scale (NIHSS), old myocardial infarction (OMI), patent foramen ovale (PFO), years old (y.o); * Welch’s *t* test, † Fisher’s exact test.

**Table 2 jcm-10-00332-t002:** Baseline patient blood test parameters of PFO-related stoke and AF-related stroke.

Blood Test Parameters		PFO-Related Stroke	AF-Related Stroke	*p*-Value
		(*n* = 5)	(*n* = 21)	
BNP	pg/mL	39.5 ± 28.1	155.4 ± 189.0	n.s *
D-dimer	ug/mL	3.7 ± 5.6	2.0 ± 2.4	n.s *
Hb	g/dL	14.4 ± 1.3	13.8 ± 2.0	n.s
HbA1c	%	5.5 ± 0.2	6.3 ± 1.3	0.02 *
LDL	mg/dL	116.0 ± 32.0	110.3 ± 42.4	n.s
HDL	mg/dL	67.2 ± 25.6	57.5 ± 17.4	n.s
TG	mg/dL	112.6 ± 55.6	125.1 ± 98.7	n.s
Cre	mg/dL	0.84 ± 0.2	1.01 ± 0.5	n.s
eGFR	ml/min/1.73 m2	73.8 ± 17.4	58.9 ± 20.6	n.s
CCR	ml/min	83.2 ± 35.5	56.5 ± 23.7	n.s
AST	IU/l	19.0 ± 4.7	23.4 ± 5.8	n.s
ALT	IU/l	15.6 ± 2.7	15.5 ± 6.4	n.s
LDH	IU/l	178.0 ± 18.4	231.0 ± 45.2	0.02
T-bil	mg/dL	0.62 ± 0.23	0.77 ± 0.32	n.s
ALP	IU/l	182.4 ± 70.8	237.4 ± 95.3	n.s

Atrial fibrillation (AF), alkaline phosphatase (ALP), alanine aminotransferase (ALT), aspartate transaminase (AST), brain natriuretic peptide (BNP), creatinine clearance (CCR) (Cockcroft–Gault equation), creatinine (Cre), estimated glomerular filtration rate (eGFR), hemoglobin (Hb), high-density lipoprotein (HDL), lactate dehydrogenase (LDH), low-density lipoprotein (LDL), patent foramen ovale (PFO), total bilirubin (T-bil), triglycerides (TG); * Welch’s *t* test.

**Table 3 jcm-10-00332-t003:** PFO characteristics, PFO attributable fraction, and estimated two years stroke recurrence rate by RoPE score.

PFO Characteristics		PFO-Related Stroke
		(*n* = 5)
PFO Size		
Small	*n*	1
Moderate	*n*	2
Large	*n*	2
Rope Score		4.8 ± 2.4
PFO-Attributable Fraction	%	36.0 ± 37.3
Estimated Two Years Stroke Recurrence Rate	%	12.2 ± 7.2

The classification of PFO size: 0 microbubbles were classified as no shunt, one to five microbubbles as small, six to 25 microbubbles as moderate, and more than 25 microbubbles as large.; patent foramen ovale (PFO), risk of paradoxical embolism (RoPE).

## Data Availability

The data presented in this study are available on request from the corresponding author.

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
