# Peer review of "Severity by National Institute of Health Stroke Scale Score and Clinical Features of Stroke Patients with Patent Foramen Ovale Stroke and Atrial Fibrillation"

_jcm, 2021, doi:10.3390/jcm10020332_

Round 1
Reviewer 1 Report
I have read the manuscript entitled „Severity by National Institute of Health Stroke Scale score and clinical features of stroke patients with patent foramen ovale stroke and atrial fibrillation”
The authors found that:
Compared with AF-related stroke patients, stroke severity were low in PFO-related stroke patients.
I have got major remarks:- my main objection concerns the small size of the study group, which does not allow for drawing reliable conclusions,- study was not register in ClinicalTrials.gov- retrospective nature of the study, lack of long-term observation- tables are absolutely unacceptable, graphically do not correspond to the requirements of the letter (they are as images, not editable tables), captions are broken - this makes a very bad impression- the discussion is brief
Thank you for submitting the following manuscript to JCM.
While the topic of manuscripy is of considerable interest, there are concerns that your observational study does not provide significant novel insight beyond the data that have been reported.
Author Response
Thank you for your reviewing our manuscript. We really appreciate your comments and we revised current manuscript according to your comments.

Reviewer 2 Report
Authors have written an interesting article about important topic. However, some things need to be clarified.
Major concern is that number of patients in PFO-related stroke group is only five. This makes results very prone to coincidence. This fact should be mentioned and discussed in article.
It seems also obvious that if patients are younger and have less risk factors (e.g. blood pressure and HbA1C) in PFO group than the srokes within this group are also not that severe. Brain can compensate disabilities better with younger people. Also propagation of original injury is likely to stay smaller with young people with less risk factors. These things should be mentioned and discussed in article.
Some other things should also be clarified:
Abstract
It should be clarified that final study population is only 26 patients (not 82).
Introduction
Lines 39-45: Are "stroke of undetermined etiology" and "cryptogenic stroke" same thing? Please clarify.
Materials and Methods
Line 118: "thrombus including moyamoya echo in LA 118 or LAA by TEE". Please clarify what this means.
Results
Table 1 has some text on left which is not shown correctly.
Table 2 has some text on left which is not shown correctly. Some lines are probably missing. I can´t find for example NIHSS points from the Table. Also some parts of the text seem to be missing. For example OMI abbreviation should be written out. "Hypertention" in table should be "Hypertension"?
Lines 211-213: Is it necessary to give percents (20.0 and 40.0%) when there are only 5 patients?
Lines 214-216: "The RoPE score was 4.8±2.4, and the average PFO attributable fraction was 36.0±37.3%, estimated 2 years stroke recurrence rate was 12.2±7.2% in this PFO population by RoPE score". Please include explanation what does PFO attributable fraction mean.
Discussion
Lines 259-263: Please clarify why recurrence rate/RoPE Score is important in this retrospective analysis.
There are also extra spaces after words, for example; lines 45 and 50. . Please correct.
Author Response
Thank you for your reviewing our manuscript. We sincerely appreciate your comments and we revised our manuscript according to your comments. Attached please find our response to your comments.

Reviewer 3 Report
I have a few suggestions which should be consider:
- Wouldn't it be better to use the Welch's t test instead of the Student's t test? Did all variables satisfy the requirements for using the t-test? Did the authors use a nonparametric test for any variable?
- Results section should be divided into subsections for greater clarity. At present the Authors extracted only one subsection i.e. “3.1. Study population and patient features”
- Categorically the tables should be redone, just in Word. In the current form table 2A looks to be unfinished. Besides, use consecutive numbers, not 2A, 2B.
- It would be better to transform table 1 into figure, i.e. these data could be graphically presented.
- Please discuss limitations of your study at the end of Discussion section, e.g. one of the limitations is low number of patients included.
- Have the Authors performed power analysis to justified the sample size?
Author Response
Thank you for your comments for our manuscripts. We sincerely appreciate your comments to our manuscripts. We revised our manuscript according to your comments. Attached please find our response and revised manuscript.

Round 2
Reviewer 1 Report
I thank the authors for the corrections made.
I accept the manuscript in presented form.
Author Response
Reviewer #1
General Comments: Comments and Suggestions for Authors
I thank the authors for the corrections made. I accept the manuscript in presented form.
Response: We deeply appreciate your time. We would like to express our gratitude for reviewing our manuscript.
Reviewer 2 Report
Authors have improved the quality of presentation. However, this cannot correct the number of patients included in study groups.
If paper is accepted authors should remove mention about 82 patients from abstract (lines 24 and 31). It is confusing.
It should be also mentioned clearly already in abstract that this is pilot-study and groups are small.
Figure 1. "Af-related" should be "AF-related". Please correct.
Line 308: There is on extra "-" in the beginning of sentence. Please remove it.
Author Response
Reviewer #2
Comment 1: Comments and Suggestions for Authors
Authors have improved the quality of presentation. However, this cannot correct the number of patients included in study groups. If paper is accepted authors should remove mention about 82 patients from abstract (lines 24 and 31). It is confusing.
Response 1: Thank you for highlighting this issue. Where necessary the appropriate qualifiers have been inserted to avoid suggesting that the statement is established in the literature as followings; (Line23, Line30)
We delete the phrase: ‘82 patients’. We have revised as followings:(Line23: Twenty six, Line30; Of the 26 cases)
Comment 2: It should be also mentioned clearly already in abstract that this is pilot-study and groups are small.
Response 2: Thank you for highlighting this issue. Where necessary the appropriate qualifiers have been inserted to avoid suggesting that the statement is established in the literature as followings; (Line28)
We have revised as followings:(Line28: This study was designed as a single-center, small population pilot-study.). We delete Line24: Omori Red Cross Hospital.
Comment 3: Figure 1. "Af-related" should be "AF-related". Please correct.
Response 3: Thank you for highlighting this issue. Where necessary the appropriate qualifiers have been inserted to avoid suggesting that the statement is established in the literature as followings; (Figure 1)
We have revised as followings:(Figure 1)
Comment 4: Line 308: There is on extra "-" in the beginning of sentence. Please remove it.
Response 4: Thank you for highlighting this issue. Where necessary the appropriate qualifiers have been inserted to avoid suggesting that the statement is established in the literature as followings; (Line205, 216)
We have revised as followings:(Line205, 216)
Is the part you pointed out the part of Line205 (This study has some limitation ......) with (Track Changes Word function: all change history turned off)?
When (all change history) was set to On, extra(-) certainly appeared, and we managed to erase the (-) in the same part. However below the next head of sentence (There were 18 (22%) …..), It appears at the beginning of the sentence when (Track Changes Word function : All history) is turned on again.
No extra (-) or Space was entered, and it seems that it was due to Track Changes Word Function, but we tried it, but we could not fix it well.
If the part you pointed out is different, we are very sorry, but could you please point out again including the beginning of the sentence word?
Reviewer 3 Report
The paper was improved. Congratulations
Author Response
Reviewer #3
General Comments: Comments and Suggestions for Authors
The paper was improved. Congratulations
Response: We deeply appreciate your time. We would like to express our gratitude for reviewing our manuscript.